# Predicting Toxicities and Survival Outcomes in De Novo Metastatic Hormone-Sensitive Prostate Cancer Using Clinical Features, Routine Blood Tests and Their Early Variations

**DOI:** 10.3390/cancers17233806

**Published:** 2025-11-27

**Authors:** Giuseppe Salfi, Martino Pedrani, Amos Colombo, Lorenzo Ruinelli, Daniele Brenna, Chiara Maria Agrippina Clerici, Giovanna Pecoraro, Sara Merler, Caroline-Claudia Erhart, Marialuisa Puglisi, Fabio Turco, Luigi Tortola, Ursula Vogl, Silke Gillessen, Ricardo Pereira Mestre

**Affiliations:** 1Oncology Institute of Southern Switzerland (IOSI), Ente Ospedaliero Cantonale (EOC), 6500 Bellinzona, Switzerlandmartino.pedrani@eoc.ch (M.P.); daniele.brenna@eoc.ch (D.B.);; 2Institute of Oncology Research (IOR), 6500 Bellinzona, Switzerland; 3ICT (Information and Communication Technology), Ente Ospedaliero Cantonale (EOC), 6500 Bellinzona, Switzerland; 4CTU (Clinical Trial Unit), Ente Ospedaliero Cantonale (EOC), 6500 Bellinzona, Switzerland; 5Department of Oncology, University of Naples Federico II, 80138 Naples, Italy; 6Oncology Department, San Maurizio Central Hospital of Bolzano, South Tyrolean Health Service, 39100 Bolzano, Italy; 7Faculty of Biomedical Sciences, Università della Svizzera Italiana, 6900 Lugano, Switzerland; 8Department of Human Pathology “G. Barresi”, School of Specialization in Medical Oncology, University of Messina, 98125 Messina, Italy

**Keywords:** prostate cancer, hormone-sensitive, machine learning, prognostic, blood tests, toxicity, mHSPC, ARPI, adverse events

## Abstract

In metastatic hormone-sensitive prostate cancer (mHSPC), prognostic stratification still relies on a limited set of clinical factors, mostly collected before treatment. With 7-month PSA emerging as a prognostic marker, our retrospective study aimed at investigating whether routinely collected laboratory and vital sign data, monitored during the same early treatment period, could identify significant toxicities; whether these toxicities could be predicted using baseline variables; and whether early variations in these parameters could be employed to improve outcome prediction. We confirmed, among 363 patients with de novo mHSPC, that we could automatically detect early adverse events from electronic medical records, that these correlated with survival outcomes in Cox analyses, and that their detection improved a machine-learning-based prediction of poor progression-free survival, beyond standard clinical prognostic factors. Early monitoring of routinely collected data can therefore serve as a tool for prognostic stratification in mHSPC.

## 1. Introduction

Prostate cancer (PC) is the second most common solid cancer in men [1] and is associated with a favorable prognosis in localized forms. However, in the case of metastatic PC, the five-year survival rate drops significantly to approximately 32% [2]. Novel standard-of-care options for metastatic hormone-sensitive PC (mHSPC), including combinations of androgen deprivation therapy (ADT), androgen receptor pathway inhibitors (ARPI), and/or docetaxel, have significantly improved patient prognosis [3], but a subgroup of these patients still show an aggressive and rapidly progressive disease course, highlighting the need to accurately predict disease progression and treatment response for optimal individualized patient care.

In patients with mHSPC, several clinical and demographic factors may influence prognosis and/or help to predict treatment response. These prognostic factors are typically assessed at diagnosis and guide clinicians in predicting disease trajectory and tailoring treatment. Specifically, disease burden (first introduced in the CHARTEED trial as disease “volume” [4]) correlates with survival outcomes and benefit from specific treatments [5,6]. Moreover, patients with de novo mHSPC exhibit worse outcomes compared to those with metachronous metastases [7]. Finally, younger age (<75 years old) [8] and higher BMI [9] have also been associated with improved overall survival (OS) and higher benefit from ARPIs [10].

Further prognostic insights can be gleaned during the early phases of systemic treatment. For instance, PSA levels ≤ 0.2 ng/mL after seven months of ADT are predictive of improved outcomes, as shown in the SWOG9346 [11] and CHARTEED [12] studies. Similar findings have been reported for PSA ≤ 0.1 ng/mL at six months in LATITUDE [13] and for deep PSA declines (reduction of ≥90% from baseline or to ≤0.2 ng/mL) in patients receiving ADT plus apalutamide in TITAN [14].

However, further dynamic changes occur during the first months of treatment for mHSPC, including changes in blood tests and vital parameters. Beyond PSA dynamics, early changes in physiological and laboratory parameters, including those reflected by treatment-related adverse events (AEs), may provide further insight into patient biology. These AEs, often related to systemic hormonal and chemotherapeutic impact, include fatigue, anemia, cardiovascular effects and more [15,16,17]. ARPIs further enhance the risk of cardiovascular and neurological AEs and are also associated with the risk of developing electrolyte disturbances, increased transaminase levels, and skin reactions [18,19,20,21,22,23]. Hematological and mucosal toxicities, alongside neuropathies, are linked with docetaxel treatment [4,24].

The variability and underreporting of early AEs onset across trials complicate efforts to both predict their occurrence and to assess their potential prognostic relevance. However, the advent of fully digitalized electronic medical records has made it easier to systematically capture real-world patient data and monitor even subtle changes in vital signs, laboratory results, and biometrics.

The integration of machine learning (ML) models into real-world cancer research enables efficient analysis of high-dimensional data, transforming it into actionable clinical insights [25]. This approach has been explored in various aspects of PC diagnosis and management [26], and was recently employed in a groundbreaking integrated way to predict treatment response in patients treated with immune checkpoint inhibitors in clinical trials for all cancers, through the SCORPIO ML system [27].

To our knowledge, this study is the first to systematically evaluate whether dynamic changes in routine laboratory markers and vital signs during the first seven months of treatment in de novo mHSPC can: (1) predict or anticipate the onset of organ-specific toxicities and (2) refine prognostic predictions beyond established baseline clinical parameters. We estimate risk and incidence of clinically meaningful parameter shifts and assess their prognostic value using both traditional Cox models and machine learning algorithms, with the aim of improving risk stratification and informing therapeutic decision-making.

## 2. Materials and Methods

This is a single-center retrospective study analyzing data extracted from electronic medical records of patients who consecutively presented with de novo mHSPC and initiated first-line therapy at the Oncology Institute of Southern Switzerland between the years 2014 and 2023.

Every patient either signed an informed consent for data collection or was deceased at the time of data analysis. The study was in the scope of a retrospective data collection protocol approved by the local ethics committee and was conducted in accordance with the 1964 Helsinki Declaration and its later amendments or comparable ethical standards.

A comprehensive summary of the methodologies employed is presented in Figure 1.

We included patients with de novo metastatic PC who received first-line treatment with either ADT monotherapy or ADT in association with docetaxel and/or an ARPI, with available data on their laboratory test results at diagnosis and within seven months from treatment start. Detailed inclusion and exclusion criteria can be found in Appendix A.

### 2.1. Clinical Measurements and Variable Definition

To identify factors potentially influencing survival outcomes in Cox analyses and ML models and to detail the clinical changes happening within the early phases of systemic treatment for mHSPC, we collected a comprehensive set of demographics, clinical, and laboratory parameters, as detailed in Appendix A. Patient clinical and laboratory data were collected from treatment initiation to seven months after ADT start.

Early changes in the collected vital and blood test parameters were automatically detected and classified by our IT platform based on the Common Terminology Criteria for Adverse Events version 5.0 (CTCAE v5.0), which grades events on a severity scale from Grade (G) 1 to G4. G0 defined values within the normal range. Values not classified as events under CTCAE v5.0 were categorized as either normal (G0) or abnormal (>G0), depending on whether they fell within or outside the normal ranges.

For certain variables (e.g., hypertension), we implemented a modified CTCAE v5.0 grading system. This adaptation allowed for streamlined, automated grading based solely on objective clinical and laboratory measures, while excluding clinical insights and patient-reported severity, which were not consistently available across all patients. Variables associated with broader AE categories (i.e., hematological toxicity, liver toxicity, kidney-related toxicity, electrolyte disturbances) were further grouped to facilitate comprehensive reporting and robust evaluation of their potential prognostic impact.

When Cox regression or machine learning analyses required comparisons between two groups based on the same variable, we considered G0 versus G1–4 (event did not occur vs. any-grade event occurred).

A detailed description of early monitoring-derived variables, their grading system, and monitoring procedures is provided in Appendix A.

### 2.2. Study End Points and Outcomes

Our primary objectives were to compare Progression-Free Survival (PFS) and OS between patients that, within seven months from hormonal treatment initiation, did or did not develop the selected early changes in vital parameters or blood exams. Additionally, we aimed to develop multiple ML models that further integrated established clinical prognostic data with the variables generated by the monitoring of outpatient vital signs and blood exam results collected within the first 7 months of treatment initiation. These models were designed to predict the occurrence of a PSA reduction to a value of <0.2 ng/mL within the first 7 months of treatment, as well as to predict PFS and OS outcomes. Specifically, we aimed to predict the probability of each patient falling into the best quartile or the worst quartile of PFS and OS (Appendix A).

Secondary objectives were to compare the incidence of the selected early changes in vital parameters or blood exams between different clinically-defined subgroups of our patient population and assess the performance of ML algorithms in integrating baseline demographic, clinical, and blood exam data to predict the development of early vital parameters and blood exam changes or adverse events.

The decision to assess variable changes within seven months from treatment initiation was based on the proven reliability of the same time point in assessing treatment response and prognosis for mHSPC patients, as demonstrated by the well-established use of PSA measurements at seven months as a validated surrogate marker [12].

The employed definitions of 7-month PSA, PFS, and OS [28,29] are provided in Appendix A.

Data sources and extraction process are described in Appendix A.

### 2.3. Statistical Analyses

We employed Pearson’s chi-square test to examine associations between categorical variables, comparing patients that received different mHSPC treatment regimens (ADT monotherapy, ADT + ARPI, and ADT + docetaxel ± ARPI). The association between variables and OS or PFS was depicted using Kaplan–Meier survival curves, and group comparisons were made using the log-rank test. Patients without a documented event were censored at their last follow-up. All statistical comparisons were made with two-tailed tests.

We performed univariable analyses using Cox proportional hazards models, incorporating early monitoring-derived variables (Appendix A). Variables showing significance (*p* < 0.10) in univariable analyses were included in the multivariable Cox model. This less restrictive significance level is a conventional approach for variable selection in prognostic models, aimed at preventing the premature exclusion of potentially important covariates. To address immortal time bias, we conducted landmark analyses at 6 and 7 months from the start of treatment for both PFS and OS. Cox regression analyses should be deemed exploratory; their *p*-values are not corrected for multiple testing and should be interpreted accordingly. The results are presented as hazard ratios (HR) with 95% confidence intervals (CI95%). All statistical analyses were carried out using Jamovi software version 2.7.6. Detailed statistical procedure is provided in Appendix A.

### 2.4. Model Development and Evaluation

Data preprocessing is detailed in Appendix A. To adjust for censoring in the dataset, a Kaplan-Meier estimator was used to compute inverse probability of censoring weights (IPCW).

Three machine learning models were chosen for training: Light Gradient Boosting Machine (LGBM) Classifier [30], Support Vector Classifier (SVC), and Random Forest (for both SVC and Random forest models, the Scikit-learn library [31]). These models were selected due to their complementary strengths in handling diverse types of data and their ability to capture complex patterns: LGBMClassifier: Effective in handling large datasets with missing values and features of varying importance; SVC: Robust in high-dimensional spaces and effective in separating non-linear relationships; Random Forest: Known for its interpretability and capacity to handle overfitting via ensemble learning.

The models were trained using GridSearch with Cross-Validation (from Scikit-learn library [31]) to optimize hyperparameters and improve model stability. 10-Fold Stratified Cross-Validation was utilized, ensuring that the proportion of classes was preserved across folds. This method provided robust estimates of model performance and minimized biases introduced by class imbalance.

After completing the training and hyperparameter tuning process, the models were evaluated on an unseen test set, which comprised 20% of the original dataset. The performance of the models was assessed using the area under the Receiver Operating Characteristic (ROC-AUC) curve, the Precision-Recall (PR-AUC) curve, and the Brier score to evaluate probabilistic accuracy. To ensure the reliability of Brier scores, all models were calibrated prior to evaluation.

Missing data handling is detailed in Appendix A.

The description of the calibration process is available in Appendix A. To ensure reliable probabilistic predictions, all models were calibrated using scikit-learn’s CalibratedClassifierCV with isotonic regression. We used a 5-fold cross-validation within the calibration process to help prevent overfitting and ensure robust generalization of calibrated probabilities to unseen data.

The complete dataset will be provided upon reasonable request (Appendix A).

## 3. Results

### 3.1. Study Population Characteristics

During the study period, a total of 363 patients with de novo mHSPC were included with a median follow-up of 69.5 months. Table 1 presents the baseline demographic and clinical characteristics at the time of treatment initiation (T0), distinguishing between patients treated with ADT, ADT + ARPI, and docetaxel-based regimens (either ADT + docetaxel or ADT + ARPI + docetaxel). Among the patient cohort, 245 (67.5%) received ADT monotherapy, 82 (22.6%) received ADT + ARPI, and 36 (9.9%) were treated with a Docetaxel-based combination (of which 11 received ADT + docetaxel + one ARPI). As expected, patients receiving ADT + docetaxel ± ARPI had higher rates of high-volume disease according to CHAARTED criteria (*p* < 0.01) and younger median age at diagnosis (*p* < 0.01). The relatively high number of patients treated with ADT alone reflects the historical nature of the dataset, which includes patients diagnosed as early as 2014, a period preceding the widespread adoption of ARPIs and consistent with the scientific evidence available at that time.

### 3.2. Incidence of Early-Onset AEs According to the Treatment Regimen

Appendix A presents a comparison of the incidence of events, defined by vitals or blood exam changes within the first seven months of systemic treatment, between patients who were treated with different drug regimens.

Data availability is reported in the table. Notably, it was possible to assess the incidence of hematological toxicities, electrolyte imbalances, and liver- and kidney-specific toxicities in approximately 70% of the patient cohort. For all defined events, except for the development of hypo-/hypermagnesemia and the elevation in blood GGT levels, data were available for at least 50% of the patients.

A higher incidence of hematological toxicities (*p* < 0.01), specifically neutropenia (*p* < 0.01) and anemia (*p* < 0.01), was observed in the docetaxel-treated group, while electrolyte disturbances (*p* = 0.87), as well as liver (*p* = 0.36) and kidney toxicity (*p* = 0.30) did not show significant differences between treatment groups. Patients receiving docetaxel further experienced higher incidence of decreased albumin levels (*p* = 0.03) and increased creatinine (*p* = 0.01) during the first seven months of treatment.

### 3.3. Integrating Baseline Clinical Factors with Baseline Blood Exams and Vitals Data to Predict Early Onset Biochemical Alterations

Appendix A summarizes the performance of LightGBM models to predict the development of any of the early monitoring events of the prespecified categories (hematological toxicity, liver toxicity, kidney-related toxicity, electrolyte disturbances, any abnormality) using two different data sources: one containing only the traditional baseline prognostic clinical data (baseline clinical model) and the other including also baseline vital parameters and laboratory values (baseline combined model). Overall, the addition of vitals and blood test data to the model allowed for a higher performance (compared to clinical data alone) in predicting the development of any significant biochemical alteration within 7 months from ADT start (AUC = 0.71 vs. 0.59).

However, both clinical and combined models failed in predicting organ-specific toxicities with very limited AUC improvements (AUC < 0.70) (Appendix A).

### 3.4. Impact of Early-Onset Biochemical Alterations on Survival Outcomes

In univariate Cox regression analysis (Table 2), patients who developed any-grade hematological toxicity within the first seven months of treatment had significantly shorter survival outcomes compared to those who did not. Specifically, this was associated with worse PFS (median: 12.4 vs. 23.4 months; HR: 1.56, 95% CI: 1.18–2.08; *p* = 0.0018) and OS (median: 23.2 vs. 43.7 months; HR: 1.72, 95% CI: 1.27–2.34; *p* < 0.001). Similarly, the development of electrolyte disturbances was associated with worse survival outcomes, including shorter PFS (median: 15.4 vs. 22.8 months; HR: 1.47, 95% CI: 1.09–1.99; *p* = 0.011) and OS (median: 26.7 vs. 44.4 months; HR: 1.73, 95% CI: 1.24–2.40; *p* = 0.001). Finally, liver-related toxicity was associated with a reduced PFS (HR: 1.31, 95% CI: 0.98–1.75; *p* = 0.073), but its association with OS was not statistically significant (HR: 1.30, 95% CI: 0.95–1.78; *p* = 0.099). Kidney-related toxicities were not significantly associated with PFS (HR: 1.28, 95% CI: 0.90–1.82; *p* = 0.174), but their development was predictive of a shorter OS (median: 25.9 vs. 41.5 months; HR: 1.60, 95% CI: 1.11–2.30; *p* = 0.012).

In the multivariable OS cox analysis (Appendix A), ALP level increase (HR: 1.93, 95% CI: 1.18–3.17; *p* = 0.009), decreased albumin levels (HR: 1.92, 95% CI: 1.18–3.13; *p* = 0.008), and the development of hyponatremia (HR: 1.79, 95% CI: 1.05–3.04; *p* = 0.033) remained independent prognostic factors for worse survival. Conversely, achieving a 7-month PSA < 0.2 ng/mL was strongly associated with longer survival (HR: 0.15, 95% CI: 0.07–0.30; *p* < 0.001). Kaplan-Meier curves describing PFS and OS in patients experiencing changes in selected biochemical parameters within the first 7 months of treatment are shown in Appendix A, and the results of multivariable analyses are detailed in Appendix A.

### 3.5. Integrating Clinical Prognostic Factors with Early Monitoring Data into ML Models to Predict Survival Outcomes

Three ML models—LGBM Classifier, SVC, and Random Forest (RF)—were developed to predict the probability of each patient to experience a 7-month PSA decrease to values below 0.2 ng/mL, as well as the probability of falling within the best or worst quartiles of PFS and OS based on the general population’s distribution. The performance of these models, evaluated on three different data sources (Clinical, Monitoring, and Combined), is summarized in Table 3.

Dynamic models integrating clinical features with longitudinal laboratory and vital sign monitoring data demonstrated superior performance in predicting shorter PFS (bottom quartile) compared with models relying solely on clinical variables. Specifically, models based on the combined dataset achieved ROC–AUC values ranging from 0.91 to 0.94, versus 0.79–0.89 for the clinical-only datasets, with consistently low Brier scores (0.12–0.14).

For the prediction of longer OS (top quartile), models incorporating monitoring data achieved good discriminative ability, with the monitoring-only dataset yielding the best results (ROC–AUC = 0.79, 0.83, and 0.80 for LGBM, SVC, and RF, respectively). The combined dataset further improved performance relative to clinical-only models (ROC–AUC range = 0.69–0.86 vs. 0.61–0.71).

Conversely, the predictive performance of models for 7-month PSA response, higher PFS (top quartile), and lower OS (bottom quartile) was substantially lower across all algorithms and data sources, indicating limited discriminative ability in these subsets. Nonetheless, models trained on the combined dataset maintained low Brier scores (<0.20) across all tasks, except for those predicting 7-month PSA response.

SHAP summary plots for selected models are presented in Appendix A. Across models predicting PFS, features with the largest contributions included liver and hematologic toxicities, electrolyte disturbances, increased serum ALP, and decreased albumin, together with ISUP grade group and 7-month PSA. For OS, the models highlighted similar contributors, with ISUP grade group, electrolyte disturbances, increased ALP, liver toxicity, and elevated CRP showing the strongest impact on predicted shorter survival.

Appendix A shows representative patients from the cohort with different baseline characteristics and outcomes. Each feature’s contribution varied in direction and magnitude based on its value and the values of other features, demonstrating the model’s complexity in predicting treatment response and survival for each patient.

## 4. Discussion

To the best of our knowledge, this is the first study to comprehensively assess early variations in vital parameters and blood test results during systemic treatment in a real-world cohort of patients with de novo mHSPC. It also evaluates their prognostic role using both classical statistical methods and ML systems. Finally, we present the first prognostic ML model based on established prostate cancer-specific prognostic features, incorporating a comprehensive view of vital parameters, blood tests, and their dynamic modifications within the first seven months of treatment.

Changes in key blood tests, such as PSA, within the first seven months of treatment are well-established prognostic biomarkers for these patients [12]. While we hypothesized that other dynamic changes in non-cancer-specific parameters might offer additional prognostic value, possibly also reflecting treatment-related toxicities, we acknowledge the valid concern that such associations could represent statistical correlations rather than direct biological causality. We therefore selected this specific timeframe to develop an automated model that detects and grades deviations from baseline in vital signs and blood test values, recognizing that the ability of machine learning to identify complex relationships is key to uncovering potentially non-intuitive prognostic signals.

We developed the first ML model, trying to offer an individualized risk assessment for the probability of developing some specific AEs (hematological, kidney, hepatic and electrolyte disorders) defined by variations of blood tests during the first seven months of treatment. In this context, however, when looking at each single organ-specific AE, models that used both clinical and vitals/laboratory variables showed reduced prediction accuracy compared to those that employed only clinical and demographic factors. Overall, the ML models’ performance was low with both training datasets. These findings highlight the intrinsic complexity of predicting organ-specific toxicities and suggest that unmeasured biological, clinical, or treatment-related factors likely contribute to these events beyond the variables captured in our dataset. They also underscore the need for a more refined selection of baseline features to improve future toxicity prediction efforts.

Our univariable Cox analysis revealed that the development of hematological, liver and kidney-related toxicity, as well as the development of electrolyte disturbances within the first seven months of systemic treatment for mHSPC, was associated with significantly shorter PFS and/or OS.

As patients receiving docetaxel had higher rates of hematological toxicities, the observed association with survival outcomes may reflect the selection of patients with more widespread and aggressive PC.

To account for such potential confounders, we looked at single factors correlating with survival outcomes in multivariable Cox analysis that also considered 7-month PSA levels. Ultimately, only increasing ALP levels, decreasing albumin, and hyponatremia development were associated with a shorter OS.

Within published studies, elevated serum ALP represents a biomarker for bone disease progression with an established role in predicting PC progression and patient survival [32,33,34]. Similarly, low serum albumin values are an established negative prognostic factor in patients with cancer [35,36] and PC specifically [37,38], underscoring the reliability of our dataset to capture significant factors influencing disease progression and survival outcomes. When looking at other blood test values, a limited amount of data is available regarding the predictive and prognostic role of their baseline value and their variation during hormonal treatments for mHSPC. For hyponatremia development, its role in prostate cancer is understudied. Evidence from other malignancies suggests that hyponatremia may reflect more aggressive tumor biology or uncontrolled disease growth, but its precise significance in de novo mHSPC remains unclear. In the Swedish AMORIS study [39] baseline sodium values did not correlate with PC mortality. In that study, patients with localized PC were mostly included. It is reasonable to assume that the development of hyponatremia in the early phases of treatment for mHSPC represents a distinct phenomenon, which we showed has significant prognostic value in our cohort. However, that is likely to represent a marker of adverse disease biology or treatment-related stress rather than a causal driver of poor survival.

Our study also explored the potential prognostic and predictive value of significant variations in vital signs during the early phase of treatment. However, the availability of these parameters from outpatient visits was highly limited, leading to substantial missing data that weakened their statistical power and precluded any meaningful association with treatment response or survival outcomes. In a recent study, we had previously demonstrated that the development of clinically significant hypertension was an independent predictive and prognostic factor in a real-world population with de novo mHSPC [40]. However, in that study the grading system strictly followed the CTCAE v5.0 system, which included and mainly relied on the introduction of new antihypertensive drugs. In the present study, we could not account for such information in order to allow the IT system to generate an automated grading system based on outpatient blood pressure measurements only, and this adapted definition of hypertensive toxicity did not show significant correlations with survival outcomes. This highlights the need to capture hypertensive drug introduction to better identify this frequent AE development and exploit its prognostic and predictive role. Given the extent of missing data in our cohort, the impact of vital sign fluctuations warrants a dedicated prospective investigation in future studies.

Finally, in this study, we provide an mHSPC-specific ML platform to predict patient outcomes. Our model integrates well-established clinical predictive and prognostic factors, such as age, Gleason Score, disease burden, and PSA, with a wider range of vital parameters, blood test results, and patient characteristics, allowing for a comprehensive characterization of each patient. Furthermore, we account for the described dynamic changes that occur during the first seven months of systemic treatments. By incorporating all these features into our ML models we could generate individual prognostic scores for every patient. This results in the first-of-their-kind comprehensive ML-based prognostic models for mHSPC, demonstrating promising results.

Notably, models incorporating changes in blood tests and vital parameters recorded during the first seven months of treatment exhibited significantly higher accuracy in predicting the probability of patients to be within the bottom quartile of PFS, compared to models based solely on prognostic baseline clinical features. SHAP summary plots from our ML models indicate that the early emergence of hematologic and hepatic toxicities, electrolyte abnormalities, rising serum ALP, and declining albumin levels contribute additional prognostic information beyond established baseline predictors of poor outcome. These dynamic on-treatment changes appear to reinforce, and in some cases amplify, the prognostic signals identified in our traditional statistical analyses.

Our findings suggest that monitoring vital parameters and blood test results during the first seven months of treatment for mHSPC provides significant predictive information and plays a critical role in identifying patients at higher risk for poor survival outcomes, while clinical data remain strong indicators of favourable prognosis. Overall, our ML models employing dynamic monitoring-derived variables consistently demonstrated an improved performance in prognosticating 7-month PSA, PFS, and OS, underscoring the value of early monitoring in refining the prediction of clinical outcomes in mHSPC patients.

Our observations collectively suggest that predicting toxicities and parameter changes in de novo mHSPC is challenging and often imprecise, even when considering clinical data and treatment regimens. However, detecting these changes and integrating them into ML models can help improve the identification of patients at risk at very early progression and death and predict poor 7-month PSA responses. This highlights the value of real-time biochemical monitoring for the early detection of variations in vital parameters and blood test results during systemic treatments.

These results emphasize the potential of early monitoring data to improve patient outcome prediction. Therefore, the implementation of standardized and structured data collection systems within hospital settings should be encouraged. Improved data integration would enable more accurate and timely prognostic modelling, ultimately supporting more personalized treatment decisions and optimized patient management in clinical practice.

Future work should also consider integrating additional variables into these models. Tumor genomic features and derived classifiers have only recently shown for the first time their predictive applications for patients with mHSPC. The 22-gene mRNA-based Decipher genomic classifier demonstrated significant predictive value in identifying patients with de novo mHSPC who derived overall survival benefit from the addition of docetaxel to ADT [41]. The same classifier was further coupled with a PTEN mRNA score, improving the predictive ability to identify this subpopulation of STAMPEDE that benefits from docetaxel [42]. Retrospective single-center analysis suggested that testing for alterations in genes associated with the “aggressive variant” molecular subtype, when integrated with clinical factors such as the CHAARTED volume criteria, could help identify those patients most likely to benefit from the addition of ARPI to ADT [43,44], Incorporating molecular data into ML frameworks may improve the ability of these models to capture inter-patient tumor heterogeneity and enhance the accuracy of survival predictions, potentially introducing predictive applications.

A direct comparison between the performance of our models and that of established genomic classifiers like Decipher is not possible within the scope of this study. However, our approach offers a distinct potential advantage: it relies on inexpensive, serially collected, and readily available clinical data, making it highly accessible in any clinical setting. In contrast, genomic classifiers, although increasingly informative, remain limited by higher costs, variable availability across institutions, and longer turnaround times. Future studies could explore how our data-driven prognostic scores could be integrated with genomic information. This could create a multi-modal approach where our model serves as an initial, routine risk stratification tool, with more complex genomic tests reserved for specific, high-stakes clinical decision points. However, real-world feasibility and resource availability should be taken into consideration when defining research priorities in this setting.

While our models are exploratory, they raise important questions about potential changes to clinical management. If validated in prospective studies, the integration of dynamic biochemical monitoring could, for instance, lead to more frequent monitoring for patients identified as high-risk during the initial phase of therapy. Furthermore, the early detection of a high-risk trajectory could one day serve as a trigger to consider treatment intensification or enrollment in clinical trials for novel therapies, moving towards a more adaptive and personalized treatment approach.

Thus, larger studies and prospective validation within clinical trial cohorts are needed to further assess the potential of our ML models in predicting adverse events and survival outcomes in patients with mHSPC.

### Limitations and Future Perspectives

The main limitations of our study include its retrospective single-center nature, relatively small sample size, and treatment heterogeneity. The significant amount of missing data for some follow-up laboratory parameters (up to 30% for certain toxicities) is also a key limitation. This is common in retrospective real-world studies where lab tests are not performed on a fixed schedule as in clinical trials. We chose not to exclude these patients to maximize the cohort size and maintain a representative real-world population, but this may have introduced bias and reduced the power of our analyses on specific toxicities. Moreover, the reported data were limited to what was recorded during oncological outpatient visits.

A key characteristic of our cohort is the high proportion of patients (67.5%) treated with ADT monotherapy, including a majority with high-volume disease. This reflects the historical nature of our dataset, which includes patients diagnosed before the widespread adoption of upfront treatment intensification. While this allowed us to evaluate biomarkers in a heterogeneous treatment landscape, it is a significant limitation. The prognostic power of the variables identified in our study might differ in a contemporary cohort of patients uniformly treated with more effective combination therapies (e.g., ADT plus an ARPI). The natural history of the disease is altered by modern treatments, and it is plausible that the impact of certain biochemical alterations on survival is modulated by the efficacy of the systemic therapy. Therefore, the generalizability of our findings to current clinical practice is limited, and our models urgently require validation in cohorts treated according to the present-day standard of care. To partially mitigate the baseline and treatment-related differences among our patients, we applied multivariate Cox models adjusting for established prognostic factors and employed machine-learning models with feature-attribution analyses, allowing us to evaluate the independent contribution of monitoring-based events despite underlying heterogeneity.

In addition to the small sample size, the exploratory nature of our study involved numerous statistical comparisons, and we did not apply a formal correction for multiple testing. This increases the risk of Type I errors (false-positive findings), and thus the significance of individual associations should be interpreted with caution.

A further limitation of this study is the presence of missing laboratory and vital-sign data, which may have reduced statistical power for certain variables and introduced potential selection bias despite the use of multivariate adjustment and ML models capable of handling incomplete data.

Furthermore, while we explored several machine learning models to assess the robustness of our findings, applying multiple algorithms to a dataset of this size increases the risk of identifying spurious associations. These models should be considered exploratory, and their performance requires validation in larger, independent cohorts. We chose to employ a disease-specific model in a selected population of patients with de novo mHSPC to minimize possible confounding factors such as population heterogeneity, prior treatment exposure, and potential tumor biological evolution.

Lastly, while we did not account for all patient comorbidities, we attempted to minimize this bias by restricting the cohort to patients with an ECOG Performance Status of 0–1, ensuring a relatively homogeneous population in terms of baseline functional status.

## 5. Conclusions

Our findings reveal, for the first time, that the integration of established prognostic factors with the detection of significant changes in vital parameters and blood tests occurring early during systemic treatment in patients with de novo mHSPC enhances patient stratification and improves prediction of survival outcomes.

Moreover, we show for the first time the feasibility of employing these data to generate ML models to predict treatment response and survival outcomes in patients with de novo mHSPC.

Our results, despite their retrospective nature and the small sample size, should be regarded as hypothesis-generating and encourage further similar research on larger, multi-centric datasets.

## Figures and Tables

**Figure 1 cancers-17-03806-f001:**
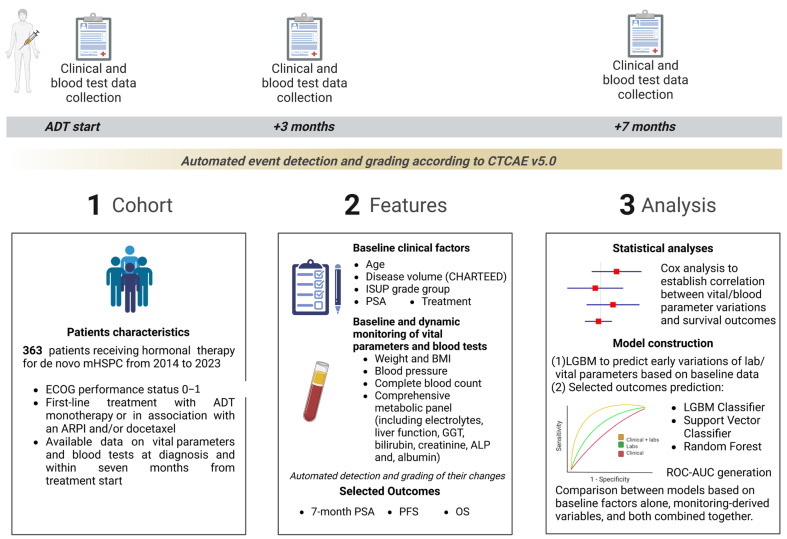
Summary of the study design and analysis. ADT, androgen deprivation therapy; ALP, Alkaline Phosphatase; ARPI, androgen receptor pathway inhibitor; BMI, body mass index; CTCAE v5.0, Common Terminology Criteria for Adverse Events version 5.0; ECOG, Eastern Cooperative Oncology Group; GGT, Gamma-glutamyltransferase; ISUP, International Society of Urological Pathology; LGBM, Light Gradient Boosting Machine; mHSPC, metastatic hormone-sensitive prostate cancer; OS, overall survival; PFS, progression-free survival; PSA, prostate-specific antigen; ROC-AUC, receiver operating characteristic and area under the receiver operating characteristic curve.

**Table 1 cancers-17-03806-t001:** Baseline clinical patient characteristics. Comparison between patients that received androgen deprivation therapy (ADT) monotherapy versus ADT plus an androgen receptor pathway inhibitor (ARPI) versus ADT plus docetaxel plus/minus an ARPI. *p*-values are calculated using Pearson’s chi-square test. Abbreviations: CI, Confidence Interval; ISUP, International Society of Urological Pathology; PSA, prostate-specific antigen; IQR, interquartile range.

Variable	Total Population	ADT	ADT + ARPI	ADT + Docetaxel ± ARPI	*p*-Value
Overall patients	363	245 (67.5%)	82 (22.6%)	36 (9.9%)	
CHAARTED					*p* < 0.01
High volume	205 (56.5%)	124 (50.6%)	51 (62.2%)	30 (83.3%)	
Low volume	158 (43.5%)	121 (49.4%)	31 (37.8%)	6 (16.7%)	
Age at diagnosis	75,	77,	76,	66,	*p* < 0.01
(median, IQR) -	67–82	69–83	67.0–81.0	63–72	
Age at diagnosis					*p* < 0.01
≥70 years old	246 (67.8%)	176 (71.8%)	57 (69.5%)	13 (36.1%)	
<70 years old	117 (32.2%)	69 (28.2%)	25 (30.5%)	23 (63.9%)	
Baseline PSA	70	100.0,	38.0,	60.0,	*p* < 0.01
(median, 95% CI)	26–330	32.0–438	18.0–156	13–209	
ISUP grade group					*p* = 0.30
5	203 (55.9%)	114 (46.5%)	63 (76.8%)	26 (72.2%)	
<5	42 (11.6%)	29 (11.8%)	9 (11.0%)	4 (11.1%)	
Unknown	118 (32.5%)	102 (41.6%)	10 (12.2%)	6 (16.7%)	
7-month PSA					*p* < 0.01
>0.2	129 (35.5%)	87 (35.5%)	29 (35.3%)	13 (36.1%)	
<0.2	72 (19.8%)	21 (85.7%)	42 (51.2%)	9 (25.0%)	
Unknown	162 (44.6%)	137 (55.9%)	11 (13.4%)	14 (38.9%)	

**Table 2 cancers-17-03806-t002:** Results of Univariable Cox Analyses of Progression-free Survival (PFS) and Overall Survival (OS) in the Overall Population. Univariable Cox regression analysis comparing patients who developed any-grade event within one of the established major categories to those who did not experience the event.

**Variable**	**Median PFS [months, 95% CI]**	**HR (95% CI)**	***p* Value**
Electrolyte disturbances			*p* = 0.011
0	22.8 [19.2–39.3]		
1	15.4 [10.1–21.3}	1.47 (1.09–1.99)	
Hematologycal toxicity			*p* = 0.02
0	23.4 [19.2–32.9]		
1	12.4 [9.7–20.2]	1.56 (1.18–2.08)	
Liver toxicity			*p* = 0.0073
0	20.5 [17–26.6]		
1	15 [10.4–22.4]	1.31 (0.98–1.75)	
Kidney-related toxicity			*p* = 0.174
0	20.5 [17–26.9]		
1	15.4 [9.7–23.4]	1.28 (0.90–1.82)	
**Variable**	**Median OS [months, 95% CI]**	**HR (95% CI)**	***p* Value**
Electrolyte disturbances			*p* = 0.001
0	44.4 [35.4–63.9]		
1	26.7 [19.8–35.7]	1.73 (1.24–2.40)	
Hematological toxicity			*p* = 0.001
0	43.7 [38–54.7]		
1	23.2 [18.5–34.4]	1.72 (1.27–2.34)	
Liver toxicity			*p* = 0.099
0	39.3 [31.9–48.1]		
1	32.1 [19.8–43.9]	1.30 (0.95–1.78)	
Kidney-related toxicity			*p* = 0.012
0	41.5 [34.4–48.1]		
1	25.9 [18.2–39.3]	1.60 (1.11–2.30)	

**Table 3 cancers-17-03806-t003:** Performance of machine learning models in predicting clinical outcomes. This table reports the performance of LightGBM (LGBM), Support Vector Classifier (SVC), and Random Forest (RF) in predicting: (1) 7 month PSA ≤ 0.2 ng/mL, and (2) membership in the best (higher 25%) or worst (lower 25%) quartiles of progression-free survival (PFS) and overall survival (OS). Models are trained on three data sources: baseline clinical variables (“Clinical”), monitoring-derived features from routine laboratory tests and vital signs (“Monitoring”), and their combination (“Combined”). Discrimination was assessed using the area under the receiver operating characteristic curve (ROC AUC), while probabilistic accuracy was evaluated using the Brier score. Higher ROC AUC indicates better discrimination; lower Brier scores indicate more accurate and better-calibrated probabilities. As a rule of thumb, Brier scores below ~0.20 generally reflect useful probabilistic performance for binary clinical predictions (lower is better), though the meaningful threshold depends on outcome prevalence and case-mix. Abbreviations: LGBM, Light Gradient Boosting Machine; SVC, Support Vector Classifier; RF, Random Forest; PSA, Prostate-Specific Antigen; PFS, Progression-Free Survival; OS, Overall Survival; ROC AUC, Area under the Receiver Operating Characteristic curve.

Target	Dataset	LGBM ROC AUC	LGBM Brier Score	SVC ROC AUC	SVC Brier Score	RF ROC AUC	RF Brier Score
PSA 7 months < 0.2	Clinical	0.62	0.24	0.54	0.26	0.58	0.27
PSA 7 months < 0.2	Monitoring	0.60	0.27	0.60	0.26	0.60	0.27
PSA 7 months < 0.2	Combined	0.68	0.24	0.63	0.26	0.67	0.25
Higher 25% PFS	Clinical	0.68	0.14	0.64	0.14	0.68	0.16
Higher 25% PFS	Monitoring	0.54	0.16	0.51	0.15	0.57	0.15
Higher 25% PFS	Combined	0.54	0.15	0.54	0.14	0.53	0.14
Lower 25% PFS	Clinical	0.89	0.19	0.79	0.18	0.86	0.16
Lower 25% PFS	Monitoring	0.80	0.17	0.84	0.13	0.85	0.15
Lower 25% PFS	Combined	0.94	0.14	0.91	0.14	0.92	0.12
Higher 25% OS	Clinical	0.68	0.12	0.61	0.12	0.71	0.14
Higher 25% OS	Monitoring	0.79	0.10	0.83	0.09	0.80	0.12
Higher 25% OS	Combined	0.69	0.12	0.80	0.11	0.86	0.11
Lower 25% OS	Clinical	0.54	0.21	0.62	0.22	0.62	0.22
Lower 25% OS	Monitoring	0.69	0.21	0.71	0.22	0.71	0.19
Lower 25% OS	Combined	0.67	0.19	0.66	0.19	0.59	0.19

## Data Availability

Individual-level patient data from the real-world datasets used in this study are not publicly available due to the number of data features drawn from the clinical testing performed as part of routine care, which could compromise the privacy of research participants. These data will be made available to researchers upon request from the corresponding author (R.P.M.) and execution of a data transfer agreement as required by the Institutional Review Boards of the authors’ institutions.

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
