# Peer review of "Predicting Toxicities and Survival Outcomes in De Novo Metastatic Hormone-Sensitive Prostate Cancer Using Clinical Features, Routine Blood Tests and Their Early Variations"

_cancers, 2025, doi:10.3390/cancers17233806_

Round 1

Reviewer 1 Report

Comments and Suggestions for Authors

This is an excellent study addressing an innovative application of machine learning to prognostication in de novo mHSPC. The results are clinically meaningful. A few aspects require clarification:

  • Missing laboratory and vital sign data are mentioned as a study limitation, but the manuscript should specify how missing values were managed in the ML workflow (e.g., imputation, variable exclusion, or listwise deletion).
    A brief description in the Methods section would improve reproducibility.
    In addition, consider adding a summary table (as an example) showing the distribution of main targets in the dataset — such as the number of patients with 7-month PSA <0.2 ng/mL and those in each quartile of PFS and OS — to give readers a clearer view of class balance.
  • Supplementary Results 6 provides useful SHAP visualizations, but the manuscript lacks a concise practical interpretation.
    Please add a short paragraph summarizing which features most consistently show a positive or negative impact on predicted poor PFS (e.g., rising ALP or declining albumin levels), to make the findings more clinically interpretable.

Author Response

Dear Editor,

We sincerely thank the Reviewer for their thoughtful feedback and insightful suggestions.

Please find below our point-by-point responses to the comments:.

Reviewer 1:

This is an excellent study addressing an innovative application of machine learning to prognostication in de novo mHSPC. The results are clinically meaningful. A few aspects require clarification:

Missing laboratory and vital sign data are mentioned as a study limitation, but the manuscript should specify how missing values were managed in the ML workflow (e.g., imputation, variable exclusion, or listwise deletion).

A brief description in the Methods section would improve reproducibility.

We thank the reviewer for this insightful comment. Missing data are indeed an important consideration in real-world cohort studies collecting longitudinal data and directly influence the design and interpretation of machine learning workflows. To address this, we employed multiple machine learning models, each with different strengths in handling diverse types of data. For instance, LGBM models were included specifically also because they can natively accommodate missing values without imputation, allowing the model to learn informative patterns directly from missingness. To improve transparency and reproducibility, we have now added a clear description of the missing-data handling procedures for both LGBM and non-LGBM models in a new Supplementary Methods section (sMethod9 - Handling of Missing Data in Machine Learning Models) of the manuscript.

In addition, consider adding a summary table (as an example) showing the distribution of main targets in the dataset — such as the number of patients with 7-month PSA <0.2 ng/mL and those in each quartile of PFS and OS — to give readers a clearer view of class balance.

We thank the reviewer for this valuable suggestion. We agree that providing a summary of the distribution of the main study targets improves clarity and helps readers better appreciate class balance in the dataset. As recommended, we have now added a summary table reporting the number of patients achieving a 7-month PSA <0.2 ng/mL, as well as the distribution of patients across quartiles of PFS and OS. This table has been included in the newly added Supplementary Results (sResults7) section.

Supplementary Results 6 provides useful SHAP visualizations, but the manuscript lacks a concise practical interpretation. Please add a short paragraph summarizing which features most consistently show a positive or negative impact on predicted poor PFS (e.g., rising ALP or declining albumin levels), to make the findings more clinically interpretable.

We thank the reviewer for this insightful comment. We agree that a concise practical interpretation of the SHAP findings enhances clinical interpretability. We have now added a short paragraph in the Results section of the manuscript. In addition, we have expanded the Discussion to comment on the potential clinical implications of these findings, noting that the SHAP-identified factors reinforce the prognostic relevance of certain emerging toxicities and laboratory dynamics that are already supported by our traditional statistical analyses.

Reviewer 2 Report

Comments and Suggestions for Authors

This manuscript presents a timely and methodologically ambitious investigation into the prognostic value of early dynamic changes in routine clinical data for patients with de novo metastatic hormone-sensitive prostate cancer (mHSPC). The application of machine learning to integrate these dynamic variables with standard baseline factors is a notable strength. The core finding—that such integration improves the identification of patients with the poorest outcomes—is novel and of potential clinical significance for risk stratification.

However, several major concerns must be addressed to strengthen the manuscript and ensure the validity and generalizability of its conclusions.

Major Concerns:

Historical Cohort and Treatment Heterogeneity: A critical limitation is the composition of the cohort, where 67.5% of patients were treated with ADT monotherapy. As this is no longer the standard of care for most de novo mHSPC patients, the generalizability of the findings to contemporary populations treated with combination therapies (ADT + ARPI/docetaxel) is severely limited. The prognostic associations identified (e.g., between hematological toxicity and survival) are likely confounded by treatment selection bias, as more aggressive diseases were likely treated with docetaxel. This fundamentally questions the applicability of the proposed model in current practice.

Substantial Missing Data and Potential Biases: The high rate of missing data for laboratory parameters (up to 30%) and vital signs is a significant weakness. This introduces a substantial risk of selection bias, as patients with complete data may systematically differ from those without. It also reduces the statistical power of the analyses. The failure to find an association with hypertensive toxicity is likely a direct consequence of both data missingness and the simplified grading system that could not account for antihypertensive medication initiation, a key component of CTCAE criteria.

Statistical and Methodological Rigor:

Multiple Testing: The study conducts numerous statistical comparisons in univariable analyzes and across multiple machine learning tasks without applying a correction for multiple testing (e.g., Bonferroni, FDR). This greatly increases the risk of Type I errors (false-positive findings). The reported p-values, especially those in the univariable analyses, should be interpreted with extreme caution, and this must be explicitly acknowledged as a limitation.

Risk of Model Overfitting: The reported exceptionally high AUC values ​​(e.g., 0.91-0.94 for predicting poor PFS) are impressive but must be viewed as exploratory. The combination of a relatively small sample size for a machine learning task, high-dimensional feature space, and testing of multiple algorithms raises serious concerns about overfitting. These models require rigorous external validation in an independent cohort before their performance can be considered reliable.

Clinical Interpretation and Causality: The study excellently identifies statistical associations but struggles to establish clinical causality. For instance, is hyponatremia a direct contributor to poor survival, or is it merely a marker of a patient's overall frailty, disease burden, or specific treatment complications? The discussion should more deeply address the challenge of interpreting these dynamic variables.

Minor Concerns:

The failure of the models to predict organ-specific toxicities, while an important finding, underscores the complexity of this task and suggests the influence of unmeasured factors.

The discussion on genomic classifiers, while interesting, highlights what is missing from the current model. A more balanced discussion on the cost-accessibility trade-offs between the proposed clinical-data model and molecular tests would be beneficial.

Comments on the Quality of English Language

 The English could be improved to more clearly express the research.

Author Response

Reviewer 2:

This manuscript presents a timely and methodologically ambitious investigation into the prognostic value of early dynamic changes in routine clinical data for patients with de novo metastatic hormone-sensitive prostate cancer (mHSPC). The application of machine learning to integrate these dynamic variables with standard baseline factors is a notable strength. The core finding—that such integration improves the identification of patients with the poorest outcomes—is novel and of potential clinical significance for risk stratification.

However, several major concerns must be addressed to strengthen the manuscript and ensure the validity and generalizability of its conclusions.

Major Concerns:

Historical Cohort and Treatment Heterogeneity: A critical limitation is the composition of the cohort, where 67.5% of patients were treated with ADT monotherapy. As this is no longer the standard of care for most de novo mHSPC patients, the generalizability of the findings to contemporary populations treated with combination therapies (ADT + ARPI/docetaxel) is severely limited. The prognostic associations identified (e.g., between hematological toxicity and survival) are likely confounded by treatment selection bias, as more aggressive diseases were likely treated with docetaxel. This fundamentally questions the applicability of the proposed model in current practice.

We thank the reviewer for their thorough evaluation and for highlighting this important limitation. We fully agree that the historical nature of our real-world cohort (data collected since 2014, during a period of evolving treatment standards) results in a predominance of patients treated with ADT monotherapy, and we acknowledge in the “Limitations and future perspectives” section of the manuscript that this treatment heterogeneity limits the generalizability of our findings to contemporary populations receiving upfront combination therapies. We also mention in the discussion section that some associations, particularly those involving hematologic toxicities, may be influenced by treatment-selection bias, as patients with more aggressive disease were more frequently treated with docetaxel.

As we recognized the issue, we tried to address it as rigorously as possible within the constraints of an observational cohort, and we adopted two complementary strategies:

  • Multivariate Cox models: We systematically adjusted for all major established prognostic baseline factors (specifically including the type of treatment received) when evaluating the prognostic role of monitoring-based events. This allowed us to assess whether these dynamic on-treatment features contributed prognostic information independently of baseline disease severity and treatment selection.
  • Machine-learning models with transparent feature-attribution analysis: Our ML workflow incorporated a broad set of clinical, laboratory, and treatment-related variables simultaneously, enabling the models to account for underlying clinical heterogeneity across patients receiving different systemic therapies. Through SHAP analysis, we could quantify the relative impact of each monitoring feature in the context of baseline clinical characteristics, thereby providing a clear assessment of how much each variable influenced outcome predictions across the entire cohort.

While we fully acknowledge and have now more prominently emphasized in the revised manuscript that treatment heterogeneity remains a significant limitation, we believe that these analytic approaches mitigate confounding to the greatest extent possible and strengthen the rationale for this hypothesis-generating study. We have expanded the “Limitations and future perspectives” section to more clearly articulate both the limitations and the interpretability advantages provided by the combined multivariate Cox and ML frameworks.

Substantial Missing Data and Potential Biases: The high rate of missing data for laboratory parameters (up to 30%) and vital signs is a significant weakness. This introduces a substantial risk of selection bias, as patients with complete data may systematically differ from those without. It also reduces the statistical power of the analyses. The failure to find an association with hypertensive toxicity is likely a direct consequence of both data missingness and the simplified grading system that could not account for antihypertensive medication initiation, a key component of CTCAE criteria.

We thank the reviewer for this important observation. Missing data are indeed an inherent frequent limitation of real-world datasets. To reduce the impact of this issue, we developed an automated framework within our institution’s electronic medical record system to systematically capture all available laboratory values for prostate cancer patients and to algorithmically grade clinically relevant changes. Despite this effort, some laboratory and vital-sign variables remained incompletely recorded, particularly those assessed less frequently in routine practice. We agree that this may have reduced power for certain endpoints and may partially explain the absence of an association with some variables (such as, as mentioned, hypertensive toxicity).

Importantly, for the majority of the variables included in our analyses, data availability was substantial enough to allow robust multivariate Cox models and machine-learning analyses. In the revised manuscript, we have expanded the description of how missing values were handled in our ML workflow, emphasizing that algorithms such as LightGBM natively accommodate missing data and therefore allow inclusion of patients with incomplete records without requiring explicit imputation, thus helping to mitigate selection bias.

Finally, we have strengthened the Limitations section to more explicitly acknowledge the potential bias introduced by missing data and the implications this may have for interpreting variables with lower completeness.

Statistical and Methodological Rigor:

Multiple Testing: The study conducts numerous statistical comparisons in univariable analyzes and across multiple machine learning tasks without applying a correction for multiple testing (e.g., Bonferroni, FDR). This greatly increases the risk of Type I errors (false-positive findings). The reported p-values, especially those in the univariable analyses, should be interpreted with extreme caution, and this must be explicitly acknowledged as a limitation.

We thank the reviewer for this important and very appropriate comment. We fully agree that the large number of statistical comparisons increases the risk of Type I error and that unadjusted p values in the univariable analyses should be interpreted with caution.

In our study, the primary analyses are the machine learning models, which are designed to capture complex, multivariable patterns associated with the outcomes. Precisely because such models can lack intuitive explainability, we included conventional univariable and Cox regression analyses with the specific, exploratory purpose of investigating and illustrating the direction and magnitude of effects for individual variables that were retained in the ML models. These classical analyses were therefore intended as hypothesis generating and as support for interpreting the more complex models, rather than as independent confirmatory tests.

To clarify this, we have now revised the Methods section to explicitly state the exploratory role of the univariable and Cox regression analyses and to note that their p values are not corrected for multiple testing and should be interpreted accordingly. In addition, we have added text to the Limitations section explicitly acknowledging the issue of multiple testing and the associated increased risk of false positive findings, particularly for the univariable results.

Risk of Model Overfitting: The reported exceptionally high AUC values (e.g., 0.91-0.94 for predicting poor PFS) are impressive but must be viewed as exploratory. The combination of a relatively small sample size for a machine learning task, high-dimensional feature space, and testing of multiple algorithms raises serious concerns about overfitting. These models require rigorous external validation in an independent cohort before their performance can be considered reliable.

We thank the reviewer for this important comment, with which we fully agree. We have extensively commented on this in both the Discussion and Limitations sections of the manuscript to emphasize that these findings are hypothesis-generating and require rigorous external validation in an independent and contemporaneous cohort before the models can be definitively considered reliable and clinically applicable.

Clinical Interpretation and Causality: The study excellently identifies statistical associations but struggles to establish clinical causality. For instance, is hyponatremia a direct contributor to poor survival, or is it merely a marker of a patient's overall frailty, disease burden, or specific treatment complications? The discussion should more deeply address the challenge of interpreting these dynamic variables.

We thank the reviewer for this valuable observation. In the revised Discussion, we have expanded our commentary on this point by more explicitly addressing the interpretative uncertainty surrounding the identified on-treatment features.

We already detailed in the manuscript that variables such as increased ALP and decreased albumin can be reasonably interpreted as reflecting underlying biological processes (bone disease activity and patient frailty, respectively) rather than acting as direct contributors to poorer outcomes. In the revised Discussion section of the manuscript we now expand on hyponatremia, which emerged as an additional monitoring-based prognostic feature in our cohort. We emphasize that its role in prostate cancer is understudied. Evidence from other malignancies suggests that hyponatremia may reflect more aggressive tumor biology or uncontrolled disease growth, but its precise significance in de novo mHSPC remains unclear. We therefore comment that hyponatremia in our cohort is more likely to represent a marker of adverse disease biology or treatment-related stress rather than a causal driver of poor survival.

Minor Concerns:

The failure of the models to predict organ-specific toxicities, while an important finding, underscores the complexity of this task and suggests the influence of unmeasured factors.

We thank the reviewer for this important point. We agree with it and we have expanded the Discussion section to address this issue more explicitly and to clarify that these negative findings further illustrate the current challenges of modelling toxicity trajectories in real-world settings.

The discussion on genomic classifiers, while interesting, highlights what is missing from the current model. A more balanced discussion on the cost-accessibility trade-offs between the proposed clinical-data model and molecular tests would be beneficial.

We fully agree with this observation, and we would like to emphasize that our models provide an easily implementable and broadly replicable framework in real-world clinical settings, as they rely on routinely collected and inexpensive biomarkers. While genomic classifiers show promise for refining prognosis and guiding treatment allocation, their higher cost and limited accessibility must be considered. In response to the reviewer’s comment, we have now balanced the Discussion section to highlight these trade-offs.

Reviewer 3 Report

Comments and Suggestions for Authors

This study aimed to comprehensively assess the clinical relevance of early dynamic alterations in vital signs and laboratory parameters within the first seven months of treatment in patients with de novo metastatic hormone-sensitive prostate cancer (mHSPC). Specifically, it investigated whether these changes could serve as early indicators of organ-specific toxicities and enhance the prediction of survival outcomes beyond conventional baseline factors.The study holds considerable clinical value; however, several limitations exist, particularly regarding study design and insufficient validation of the machine learning models. For example:
1. Although the models improved AUC values, the study did not clarify the clinical applicability of this improvement (e.g., threshold determination or potential for clinical decision support).
2. The paper did not specify how the training, validation, and test datasets were divided, nor did it mention whether external or cross-validation was performed.
3. The calculation of dynamic variables (e.g., magnitude of change, assessment intervals) was only briefly described, leaving uncertainty about whether differences in measurement timing or variability were accounted for.
4. Multiple Cox regression analyses involved numerous variables and endpoints; without adjustment for multiple comparisons, there is a potential risk of false-positive findings.
5. The study did not report whether heterogeneity existed in the predictive performance of dynamic variables among different treatment regimens (ADT alone vs. ADT+ARPI vs. ADT+Docetaxel).

Author Response

Reviewer 3:

This study aimed to comprehensively assess the clinical relevance of early dynamic alterations in vital signs and laboratory parameters within the first seven months of treatment in patients with de novo metastatic hormone-sensitive prostate cancer (mHSPC). Specifically, it investigated whether these changes could serve as early indicators of organ-specific toxicities and enhance the prediction of survival outcomes beyond conventional baseline factors. The study holds considerable clinical value; however, several limitations exist, particularly regarding study design and insufficient validation of the machine learning models. For example:

  1. Although the models improved AUC values, the study did not clarify the clinical applicability of this improvement (e.g., threshold determination or potential for clinical decision support).

We thank the reviewer for this valuable comment. Given the limited size of our single-center cohort, the primary objective of our study is hypothesis-generating: to demonstrate associations between early monitoring events, captured through routinely collected biomarkers, and patient outcomes. These associations are supported by both univariate and multivariate Cox analyses, as well as by improvements in the predictive performance of some ML models when these variables are included. Accordingly, we considered the occurrence of any grade of these events, as defined by CTCAE v5.0 (detailed in the Supplementary Materials), sufficient to establish these associations. However, due to the sample size, our study is not powered to determine precise cutoffs for individual laboratory values, and therefore we relied on the standardized CTCAE v5.0 thresholds.

  1. The paper did not specify how the training, validation, and test datasets were divided, nor did it mention whether external or cross-validation was performed.

We thank the reviewer for this valuable comment and fully acknowledge the importance of clearly reporting this data to ensure transparency and reproducibility. While we had previously attempted to provide detailed explanations in the Methods (section 2.4) and in Supplementary Methods (sMethods8–10), we recognize that relocating this information to the main text enhances clarity. Accordingly, in the revised manuscript, we have moved a portion of these details from the Supplementary Materials into the main Methods section, that we hope now could well describe the training, validation, and test dataset splits, as well as the cross-validation procedures employed.

  1. The calculation of dynamic variables (e.g., magnitude of change, assessment intervals) was only briefly described, leaving uncertainty about whether differences in measurement timing or variability were accounted for.

We thank the reviewer for this important comment. In response, we have substantially expanded the Methods section to provide a more comprehensive description of how dynamic variables were calculated, including the monitoring procedures, assessment intervals, and variable grading. Additional details are also provided in the revised Supplementary Methods (sMethod4).

  1. Multiple Cox regression analyses involved numerous variables and endpoints; without adjustment for multiple comparisons, there is a potential risk of false-positive findings.

We thank the reviewer for this important and very appropriate comment. We fully agree that the large number of statistical comparisons increases the risk of Type I error and that unadjusted p values in the univariable analyses should be interpreted with caution.

In our study, the primary analyses are the machine learning models, which are designed to capture complex, multivariable patterns associated with the outcomes. Precisely because such models can lack intuitive explainability, we included conventional univariable and Cox regression analyses with the specific, exploratory purpose of investigating and illustrating the direction and magnitude of effects for individual variables that were retained in the ML models. These classical analyses were therefore intended as hypothesis generating and as support for interpreting the more complex models, rather than as independent confirmatory tests.

To clarify this, we have now revised the Methods section to explicitly state the exploratory role of the univariable and Cox regression analyses and to note that their p values are not corrected for multiple testing and should be interpreted accordingly. In addition, we have added text to the Limitations section explicitly acknowledging the issue of multiple testing and the associated increased risk of false positive findings, particularly for the univariable results.

  1. The study did not report whether heterogeneity existed in the predictive performance of dynamic variables among different treatment regimens (ADT alone vs. ADT+ARPI vs. ADT+Docetaxel).

We thank the reviewer for this insightful comment and we agree that examining potential heterogeneity in the predictive performance of the dynamic variables across treatment regimens (ADT alone, ADT+ARPI, ADT+Docetaxel) would be of clear interest.

However, in our cohort the number of patients within each treatment subgroup is relatively limited, particularly once stratified by outcome and by the dynamic features, which substantially reduces statistical power and the stability of any subgroup specific model performance estimates. For this reason, we did not perform or report separate predictive performance analyses by treatment arm, as any apparent differences would be highly prone to random variation and difficult to interpret reliably.

At the same time, an important strength of our ML approach is that the model is designed to predict the risk of progression or death across real world treatment patterns, rather than within narrowly defined, treatment homogeneous subgroups. In other words, the model aims to provide risk estimates that are robust to the heterogeneity of systemic therapies actually used in practice, rather than targeting a single regimen in isolation.

On the other hand, we could include the type of treatment as a cofactor in our multivariate cox analysis, streghtening the robustness of the prognostic associations of some of our monitoring-derived features.

We have now clarified this points in the Methods and Limitations sections. Specifically, we state that:

  1. sample sizes within individual treatment regimens were too small to support reliable, regimen specific evaluation of predictive performance; and
  2. the scope of the current work is to model overall risk in a mixed treatment population, not to formally compare or optimize model performance separately for ADT alone, ADT+ARPI, and ADT+Docetaxel.

For future work, we agree that a pooled or larger multicenter dataset with sufficient numbers in each regimen would be the ideal setting to rigorously assess heterogeneity of model performance across specific treatment strategies.

We sincerely thank again the Reviewers for their contributions and wish that the revised version is found to be satisfactory for publication.

Kind regards

Round 2

Reviewer 1 Report

Comments and Suggestions for Authors

Thank you for your revision.

No further comment.